# Accuracy of restricted Boltzmann machines for the one-dimensional $J_1-J_2$ Heisenberg model

**Luciano Loris Viteritti[1], Francesco Ferrari[2] and Federico Becca[1]**

**1** Dipartimento di Fisica, Università di Trieste, Strada Costiera 11, I-34151 Trieste, Italy
**2** Institute for Theoretical Physics, Goethe University Frankfurt,
Max-von-Laue-Strasse 1, D-60438 Frankfurt am Main, Germany

## Abstract

Neural networks have been recently proposed as variational wave functions for quantum many-body systems [G. Carleo and M. Troyer, Science 355, 602 (2017)]. In this work, we focus on a specific architecture, known as restricted Boltzmann machine (RBM), and analyse its accuracy for the spin-1/2 $J_1-J_2$ antiferromagnetic Heisenberg model in one spatial dimension. The ground state of this model has a non-trivial sign structure, especially for $J_2/J_1 > 0.5$, forcing us to work with complex-valued RBMs. Two variational *Ansätze* are discussed: one defined through a fully complex RBM, and one in which two different real-valued networks are used to approximate modulus and phase of the wave function. In both cases, translational invariance is imposed by considering linear combinations of RBMs, giving access also to the lowest-energy excitations at fixed momentum $k$. We perform a systematic study on small clusters to evaluate the accuracy of these wave functions in comparison to exact results, providing evidence for the supremacy of the fully complex RBM. Our calculations show that this kind of *Ansätze* is very flexible and describes both gapless and gapped ground states, also capturing the incommensurate spin-spin correlations and low-energy spectrum for $J_2/J_1 > 0.5$. The RBM results are also compared to the ones obtained with Gutzwiller-projected fermionic states, often employed to describe quantum spin models [F. Ferrari, A. Parola, S. Sorella and F. Becca, Phys. Rev. B 97, 235103 (2018)]. Contrary to the latter class of variational states, the fully-connected structure of RBMs hampers the transferability of the wave function from small to large clusters, implying an increase in the computational cost with the system size.

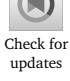
doi:[10.21468/SciPostPhys.12.5.166](10.21468/SciPostPhys.12.5.166)

# 1 Introduction

Quantum many-body systems are characterized by a Hilbert space that grows exponentially with the number of particles. This fact restricts exact calculations to a few cases, mainly for one-dimensional models, while analytical treatments often require approximations that are not fully justified in the strongly interacting limit. Therefore, numerical techniques represent a viable tool to assess the low-energy properties of these systems, beyond the perturbative regimes.

A particularly interesting class of quantum systems is represented by the frustrated spin models. Their interest relies on the possible existence of exotic phases of matter in two or three spatial dimensions, the so-called spin liquids, which are characterized by the absence of magnetic order, a high degree of entanglement, and fractional excitations, including emergent gauge fields [1]. From a numerical point of view, one difficulty in approaching frustrated spin models is related to the sign structure of the ground state, which is, in general, highly non-trivial. Consequently quantum Monte Carlo methods cannot be applied to obtain exact properties. For this reason, in the last thirty years, alternative approaches have been developed. Density-matrix renormalization group (DMRG) [2] is free from sign problems, but, while it gives excellent results for a variety of one-dimensional models, its performance considerably worsens when dealing with two-dimensional systems. In this regard, the extensions based on tensor networks (e.g., projected-entangled pair states) [3] represent a promising avenue to reach accurate results in more than one dimension, even in the thermodynamic limit. Alternatively, variational wave functions can be defined and treated within stochastic methods, without facing sign problems [4].

Various variational wave functions have been defined to deal with quantum spin models, to describe both magnetically ordered phases [5] and quantum spin liquids [6,7]. The latter ones are based on the concept of resonating-valence bond (RVB) states, which have been discussed from theoretical [8–11] and numerical [12–14] sides. In 2017, Carleo and Troyer [15] proposed a twist in the field, suggesting a new class of variational wave functions, based upon a specific class of neural networks, called restricted Boltzmann machines (RBMs). Their intrinsic correlated aspect requires numerical tools and, specifically, stochastic approaches to evaluate any observable.

One crucial advantage of neural-network states lies in the fact that they are defined by the inclusion of a set of ancillary (hidden) variables, which are coupled to the original degrees of freedom (e.g., $S = 1/2$ spins in the Heisenberg model) and whose number can be increased in order to systematically improve the quality of the variational wave function [15]. However, the number of variational parameters required by the wave functions grows polynomially with the number of hidden units. As a result, the optimization of the variational wave function

becomes a hard task due to the large number of parameters, even if stochastic approaches are employed. The original work by Carleo and Troyer was limited to Heisenberg models in one and two dimensions, where the sign structure of the ground state was known by the Marshall-sign rule [16]. This fact largely facilitates the numerical treatment, giving rise to an impressive accuracy of the neural-network states.

More complicated models, such as the frustrating Heisenberg model with both nearest-($J_1$) and next-nearest-neighbor ($J_2$) interactions, are more difficult to deal with. The main additional complication arises due to the unknown sign structure of the exact ground state, which implies the necessity of a full optimization of the variational state that involves both moduli and signs. In this regard, the difficulties in treating the sign structure have been discussed in a few works [17–20] and alternative architectures of neural networks have been devised, such as convolutional neural networks (CNNs) [18, 21–23], recurrent neural networks [24], and autoregressive neural networks [25, 26]. In addition, also combinations of neural-networks and standard variational wave functions (e.g., Gutzwiller-projected fermionic ones) have been employed [27, 28].

At present, quantum spin models on frustrated (e.g., triangular or kagome) lattices remain extraordinarily challenging problems to be addressed by numerical techniques. From one side, DMRG calculations have reached remarkable accuracies on a cylindrical geometry (with large circumferences), thus approaching the two-dimensional limit [29], also implementing clever schemes to assess the existence of highly-entangled states of matter [30]; from the other side, RBMs and, more generally, neural-network-based wave functions, which can represent quantum states in arbitrary dimensions, have been progressively improved so to reach accuracies that are comparable with state-of-the-art numerical approaches [21]. However, further improvements should be pursued, such as reducing the number of variational parameters (without losing accuracy), in order to be able to perform calculations on large clusters and assess the real nature of the exact ground state within the highly-frustrated regime, where gapped or gapless spin liquids may exist.

Here, we would like to focus on a less ambitious problem and thoroughly inspect the quality of RBM wave functions for a one-dimensional system, for which the ground state may have a highly non-trivial sign structure. In fact, the latter aspect represents an important barrier to approach generic frustrated spin models and simple test cases can provide extremely important insights. For these reasons, we consider the spin-1/2 $J_1 - J_2$ Heisenberg model on a linear chain:

$$\hat{\mathcal{H}} = J_1 \sum_R \hat{\mathbf{S}}_R \cdot \hat{\mathbf{S}}_{R+1} + J_2 \sum_R \hat{\mathbf{S}}_R \cdot \hat{\mathbf{S}}_{R+2}, \tag{1}$$

where $\hat{\mathbf{S}}_R = (\hat{S}_R^x, \hat{S}_R^y, \hat{S}_R^z)$ is the $S = 1/2$ spin operator at site $R$. Both the nearest-neighbor ($J_1$) and next-nearest-neighbor ($J_2$) couplings of the model are antiferromagnetic, i.e. $J_1 > 0$ and $J_2 \geq 0$. Our calculations are performed on finite-sized clusters with $N$ sites and periodic boundary conditions ($\hat{\mathbf{S}}_{N+1} \equiv \hat{\mathbf{S}}_1$). The ground-state phase diagram of the Hamiltonian (1) displays a gapless region, for $J_2/J_1 \lesssim 0.24$, and a gapped one, for $J_2/J_1 \gtrsim 0.24$. In the latter phase, the ground state is two fold degenerate in the thermodynamic limit, which implies a spontaneous symmetry breaking of the translational symmetry. The location of the transition point between the two phases has been computed with a very high level of accuracy by looking at the level crossing between the lowest-energy triplet and singlet excitations [31, 32]. The important aspect for the present investigation is that, while the sign structure of the ground state is rather trivial for $J_2/J_1 \leq 0.5$, it becomes highly non trivial for $J_2/J_1 > 0.5$ (see discussion below). In addition, incommensurate (spiral) spin-spin correlations are present for $J_2/J_1 \gtrsim 0.5$ [33].

We tackle the problem using two different complex-valued neural-network *Ansätze*: one written in terms of a single complex RBM, and another one in which two real-valued RBMs

are employed to separately describe the moduli and the phases of the variational state. We show that the former *Ansatz* gives a better accuracy for all the values of the frustrating ratio $J_2/J_1$ that we analysed. Particular focus is put on the ability of the RBM states to reproduce the exact sign structure of the ground state in different regimes of frustration.

## 2 Variational wave functions

### 2.1 RBM probability distribution

A class of powerful energy-based models called *restricted Boltzmann machines* (RBMs) has been widely employed in the context of machine learning to obtain accurate approximations of probability distributions [34]. Here, we give a brief introduction to this class of neural networks. Let us consider the case of a set of $N$ binary variables, which will be relevant for the quantum $S = 1/2$ Heisenberg models, $\{\sigma = (\sigma_1, \ldots, \sigma_N)\}$, distributed according to a certain probability distribution $P_0(\sigma)$. The $\sigma$-variables, dubbed *physical variables*, can take values $\pm 1$. In order to define the RBM probability distribution $P_{\mathrm{RBM}}(\sigma)$, we introduce an auxiliary set of $M$ binary (*hidden*) variables $\{h = (h_1, \ldots, h_M)\}$, which are coupled to the physical variables in the energy function [34]

$$E_{\mathrm{RBM}}(\sigma, h; \mathcal{W}) = -\sum_{i=1}^{N} a_i \sigma_i - \sum_{\mu=1}^{M} b_\mu h_\mu - \sum_{i=1}^{N}\sum_{\mu=1}^{M} \sigma_i W_{i,\mu} h_\mu. \tag{2}$$

The parameters $W_{i,\mu}$ entering the above expression are called *weights*, while $b_\mu$ and $a_i$ are the so-called *hidden* and *input biases*, respectively; the set of all parameters is denoted in a compact form as $\{\mathcal{W}\} = \{W_{i,\mu}, b_\mu, a_i\}$. The probability $P_{\mathrm{RBM}}(\sigma)$ is obtained by tracing out the hidden variables $\{h\}$ from the Boltzmann distribution of the RBM model, i.e., $P_{\mathrm{RBM}}(\sigma; \mathcal{W}) \propto \sum_{\{h\}} \exp\{-E_{\mathrm{RBM}}(\sigma, h; \mathcal{W})\}$. Due to the absence of a direct coupling between hidden variables in $E_{\mathrm{RBM}}$ (2), the trace can be performed analytically, giving:

$$P_{\mathrm{RBM}}(\sigma; \mathcal{W}) \propto \exp\left\{\sum_{i=1}^{N} a_i \sigma_i + \sum_{\mu=1}^{M} \log\left[\cosh\left(b_\mu + \sum_{i=1}^{N} W_{i,\mu} \sigma_i\right)\right]\right\}. \tag{3}$$

The result of this construction is a probability distribution function with non-trivial correlations between physical variables, parametrized by the set of weights and biases $\{\mathcal{W}\}$. For a fixed number $N$ of physical variables, the representational power of the RBM probability distribution increases with the number of hidden variables $M$ (or, equivalently, with the *complexity* parameter $\alpha = M/N$). The theoretical foundation of RBM models lies in the fact that they are universal approximators of probability distributions for sufficiently large values of $M$ [35, 36]. Indeed, by a suitable definition of a *loss function*, the parameters $\{\mathcal{W}\}$ of the RBM model can be tuned such that $P_{\mathrm{RBM}}(\sigma)$ approximates the target distribution function $P_0(\sigma)$.

### 2.2 RBM wave functions

Recently, RBMs have been used as variational wave functions to approximate the ground state of quantum many-body systems [15]. In this context, the loss function is the variational energy, which is minimized to obtain the best approximation of the exact ground-state wave function. However, contrary to probability distributions, quantum states are in general complex functions, i.e., their amplitudes in the computational basis are complex-valued. Therefore, a standard RBM parametrization making use of the $P_{\mathrm{RBM}}(\sigma; \mathcal{W}) \geq 0$ function discussed above is suitable only for those cases where the wave function is known to be real and positive definite

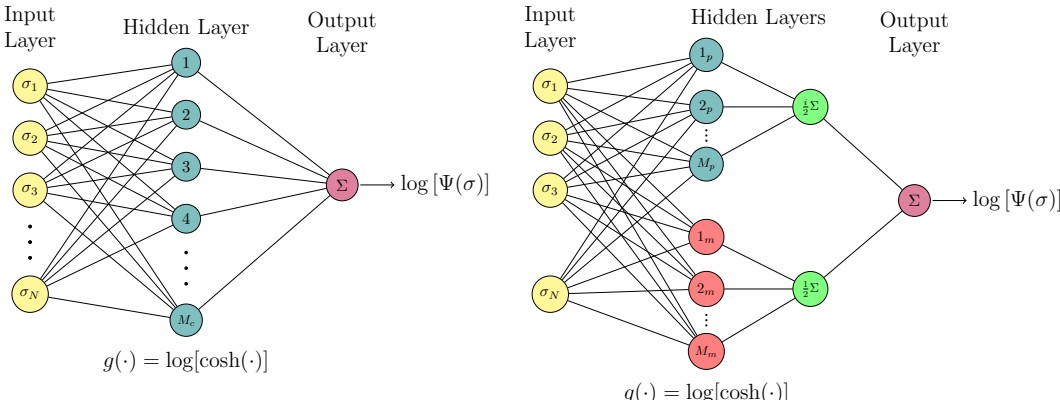

Figure 1: Schematic illustration of the feed-forward neural networks representation of the cRBM state of Eq. (5) (left panel) and pmRBM state of Eq. (4) (right panel). The cRBM *Ansatz* has complex parameters and it is a fully-connected network; instead, the pmRBM state has real parameters and, due to its structure, is not fully connected. In both networks, the activation function of the hidden neurons is $g(\cdot) = \log[\cosh(\cdot)]$.

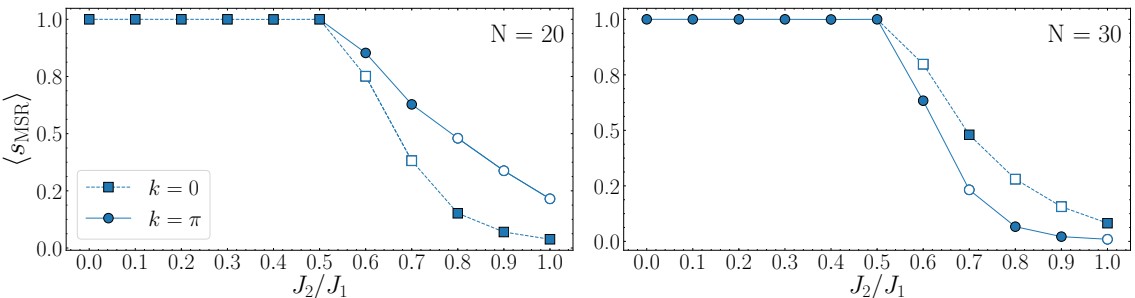

Figure 2: Average Marshall-sign defined in Eq. (6) as a function of $J_2/J_1$ for $N = 20$ (left panel) and 30 (right panel). For $J_2/J_1 < 0.5$ the ground state has momentum $k = 0$ (for $N = 20$) and $k = \pi$ (for $N = 30$); for $J_2/J_1 > 0.5$, the momentum of the ground state is not fixed. For that reason, we report both states, the actual ground state is marked by a filled symbol.

in the computational basis (e.g., in bosonic systems). For all other cases, a generalization of the above construction is required.

Within the $S = 1/2$ Heisenberg model of Eq. (1), the configurations of the physical Hilbert space can be labelled by specifying the $z$-component of the spin on each site, namely $\{|\sigma\rangle = |\sigma_1, \ldots, \sigma_N\rangle\}$, with $\sigma_i = 2S_i^z$ being a binary variable $\pm 1$. For time-reversal symmetric models, such as Eq. (1), the amplitudes of the ground-state wave function can be chosen to be real ($\langle\sigma|\Psi_0\rangle \in \mathbb{R}$), but their signs are not known in general. Representing the sign structure of the wave function with a real-valued RBM is a difficult task, which requires the treatment of non-differentiable quantities or the use of gradient-free methods for the optimization [23]. For this reason, it is often convenient to adopt a complex-valued RBM parametrization of the wave function. In this regard, two alternative formulations are presented in the following.

As a first possibility, we can employ two (independent) RBM probability functions, one for the modulus [$P_{RBM}(\sigma; \mathcal{W}_m)$] and one for the phase [$P_{RBM}(\sigma; \mathcal{W}_p)$] of the wave function [37].

The amplitudes of the quantum state are then given by:

$$\Psi_{\mathrm{pmRBM}}(\sigma; \mathcal{W}_p, \mathcal{W}_m) = \sqrt{P_{\mathrm{RBM}}(\sigma; \mathcal{W}_m)} \, \exp\left[\frac{i}{2}\Phi_{\mathrm{RBM}}(\sigma; \mathcal{W}_p)\right], \tag{4}$$

where $\Phi_{\mathrm{RBM}}(\sigma; \mathcal{W}_p) = \log[P_{\mathrm{RBM}}(\sigma; \mathcal{W}_p)]$. Here, the parameters of the RBMs, i.e., $\{\mathcal{W}_m\}$ and $\{\mathcal{W}_p\}$, are all real. The structure of the variational state is characterized by the number of hidden variables for the modulus $M_m$ and the phase $M_p$, giving the total number of hidden units being $M = M_p + M_m$. The complexity of the network is defined as the ratio between the number of hidden variables and visible ones, leading to $\alpha_m = M_m/N$ and $\alpha_p = M_p/N$. We emphasize that a different number of hidden variables can be taken for the modulus and the phase. This variational *Ansatz* is dubbed *phase-modulus RBM* (pmRBM) wave function.

The second option is taking a single RBM with complex parameters, in order to provide a complete description of both amplitude and phase of the wave function with a single complex-valued network [15]:

$$\Psi_{\mathrm{cRBM}}(\sigma; \mathcal{W}_c) = \exp\left(\sum_{i=1}^{N} a_i \sigma_i\right) \prod_{\mu=1}^{M_c} \cosh\left(b_\mu + \sum_{i=1}^{N} W_{i,\mu}\sigma_i\right). \tag{5}$$

Here, $\{\mathcal{W}_c\} \in \mathbb{C}$ and the number of hidden variables is $M_c$ corresponding to a complexity given by $\alpha_c = M_c/N$. This state is dubbed *complex RBM* (cRBM) wave function.

In the following, we set input biases equal to zero ($a_i = 0$) in both phase-modulus and complex RBMs [38–40]. Within the pmRBM state, the total number of (real) parameters is $(M_m + M_p) \times (N+1)$, i.e., $(M_m + M_p) \times N$ for the weights and $(M_m + M_p)$ for the hidden biases. Instead, the cRBM *Ansatz* contains $M_c \times (N+1)$ complex parameters, corresponding to $2M_c \times (N+1)$ real numbers, i.e., $2M_c \times N$ for the weights and $2M_c$ for the hidden biases.

In the context of machine learning, the variational wave functions defined in Eqs. (4) and (5) can be seen as feed-forward neural networks [41] with a visible layer of $N$ neurons that represent the physical configuration $\{\sigma\}$, one hidden layer of neurons with activation function $g(\cdot) = \log[\cosh(\cdot)]$, and one output neuron which performs the sum of the outputs of the hidden layer and returns the logarithm of the amplitude (see Fig. 1). The mapping between the RBM wave functions and the feed-forward neural network can be a useful starting point for possible generalizations, e.g., the so-called *n-layer feed-forward* neural network [42].

We note that for $J_2 = 0$ the lattice is bipartite and, consequently, the exact signs of the ground state wave function in our computational basis satisfy the so-called Marshall-sign rule [16], i.e., $\mathrm{sign}[\Psi_0(\sigma)] = (-1)^{N_{\uparrow,A}(\sigma)}$, where $N_{\uparrow,A}(\sigma)$ is the number of up spins on the $A$ sublattice. Motivated by this fact, we also consider variational states in which $(-1)^{N_{\uparrow,A}(\sigma)}$ is attached to the amplitudes of the RBM *Ansätze*. Although the Marshall-sign rule gives the exact signs of the ground state only in the unfrustrated limit $J_2 = 0$, it still turns out to constitute a reasonable approximation for the sign structure of the exact wave function for $J_2 \leq 0.5$ on relatively small clusters, such as the ones that can be tackled by exact diagonalization. Indeed, the accuracy of the Marshall-sign rule can be assessed by evaluating the following average:

$$\langle s_{\mathrm{MSR}} \rangle = \left| \sum_{\{\sigma\}} |\Psi_0(\sigma)|^2 \mathrm{sign}[\Psi_0(\sigma)]\mathcal{M}(\sigma) \right|, \tag{6}$$

where $\Psi_0(\sigma)$ is the exact ground-state amplitude and $\mathcal{M}(\sigma) = (-1)^{N_{\uparrow,A}(\sigma)}$ is the Marshall sign of the configuration $|\sigma\rangle$. The absolute value is taken to overcome a possible global sign in the exact state. Whenever the Marshall-sign rule is exact (e.g., for $J_2 = 0$), $\langle s_{\mathrm{MSR}} \rangle = 1$, otherwise $0 \leq \langle s_{\mathrm{MSR}} \rangle < 1$. In Fig. 2, we show the values of $\langle s_{\mathrm{MSR}} \rangle$ for $N = 20$ and $N = 30$ sites. The momentum of the ground state is either $k = 0$ or $\pi$: while for $J_2/J_1 \leq 0.5$ it does not depend

on $J_2/J_1$ but only on the parity of $N/2$, for $J_2/J_1 > 0.5$ it changes with the frustrating ratio and $N$. Therefore, for this latter case, we compute $\langle s_{\mathrm{MSR}} \rangle$ for both the lowest-energy wave functions with $k = 0$ and $\pi$. The remarkable outcome is that, even on a relatively large cluster, $\langle s_{\mathrm{MSR}} \rangle$ is very close to 1 in the whole region $0 \le J_2/J_1 \le 0.5$ (it is exactly 1 for $J_2/J_1 = 0$ and 0.5), while it rapidly drops to zero for $J_2/J_1 > 0.5$. As an example, on $N = 30$ sites, $\langle s_{\mathrm{MSR}} \rangle = 0.99994$ for $J_2/J_1 = 0.3$ and $\langle s_{\mathrm{MSR}} \rangle = 0.08195$ for $J_2/J_1 = 1$.

## 2.3 Physical symmetries

The variational wave functions discussed so far do not necessarily possess the symmetries of the physical model under investigation. In principle, the correct symmetries of the exact ground state can be potentially recovered by the variational state in the limit of a large number of hidden units, since the RBM has the property of being a universal approximator. However, in practice, we only deal with a finite number of hidden units, whose parameters are variationally optimized by numerical methods. This fact yields variational wave functions that, in general, do not fulfill the symmetries of the Hamiltonian. A possible way to overcome this issue is applying a projection operator $\hat{\mathcal{P}}_\lambda$ to enforce the desired symmetries with definite quantum numbers (denoted by $\lambda$) [43]. In general, this symmetrization procedure of the RBM states leads to a substantial improvement in the accuracy of the variational *Ansätze* [39].

In this work, we focus on a translationally invariant model and, therefore, we enforce the translational symmetry by applying the momentum projection operator

$$\hat{\mathcal{P}}_k = \frac{1}{N} \sum_R e^{-ikR} \hat{T}_R, \tag{7}$$

to the RBM wave functions. Here, $\{\hat{T}_R\}$ is the set of translation operators corresponding to the lattice vectors $\{R\}$, $N$ is the number of translations (equal to the number of sites), and $k$ is a crystal momentum. Starting from a generic (non-symmetric) quantum state $|\Psi_{\mathcal{W}}\rangle$ (either the phase-modulus or the complex RBM wave function), we define the translationally invariant state as $\hat{\mathcal{P}}_k |\Psi_{\mathcal{W}}\rangle$, whose corresponding amplitudes are given by:

$$\Psi_k(\sigma; \mathcal{W}) = \frac{1}{N} \sum_R e^{-ikR} \Psi(\sigma_R; \mathcal{W}), \tag{8}$$

where $\Psi(\sigma_R; \mathcal{W}) = \langle \sigma | \hat{T}_R | \Psi_{\mathcal{W}} \rangle$.

The projection not only improves the accuracy of the ground state variational wave function, but also gives the possibility of approximating excited states, by imposing a momentum $k$ that differs from the one of the ground state. The symmetrization procedure for restoring translational symmetry can be straightforwardly generalized to include other abelian symmetries [39]; by contrast, the inclusion of non-abelian symmetries represents, in general, a more complicated task [44, 45].

## 3 Results

The RBM variational *Ansätze* presented in the previous section are correlated many-body states, for which an analytic treatment is not possible. Their physical properties (i.e., energy and correlation functions) can be evaluated numerically by using standard variational Monte Carlo techniques, which do not suffer from any sign problem [4]. The optimization of the variational parameters can be implemented within stochastic approaches. Here, an optimization step is made by $O(10^3)$ Monte Carlo samples, each of which consists of $O(N)$ Metropolis moves (two-spin flips); variational parameters are updated at the end of every optimization step by

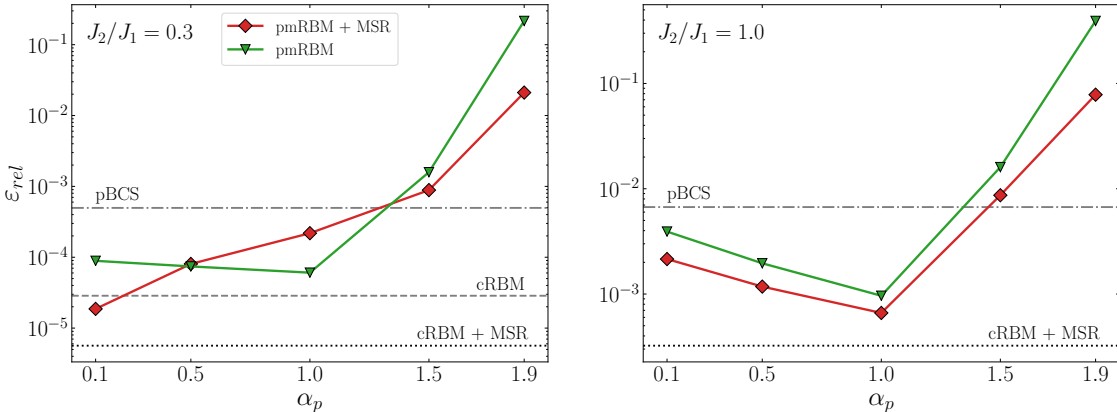

Figure 3: Accuracy of the variational energy for the $J_1 - J_2$ Heisenberg model with $N = 20$ sites, for $J_2/J_1 = 0.3$ (left panel) and $J_2/J_1 = 1$ (right panel). The pmRBM *Ansatz* of Eq. (4) is reported as a function of $\alpha_p$, with $\alpha_m + \alpha_p = 2$. The results for the cRBM wave function of Eq. (5) are reported for $\alpha_c = 1$, such that the total number of real parameters (840) is the same as for the pmRBM state. The results obtained by including the Marshall-sign rule (MSR) are also shown. For $J_2/J_1 = 1$ (right panel), the accuracy of the cRBM with and without the Marshall-sign rule do not differ, thus we included only the former one in the plot. In both panels the results obtained by pBCS states are shown for comparison.

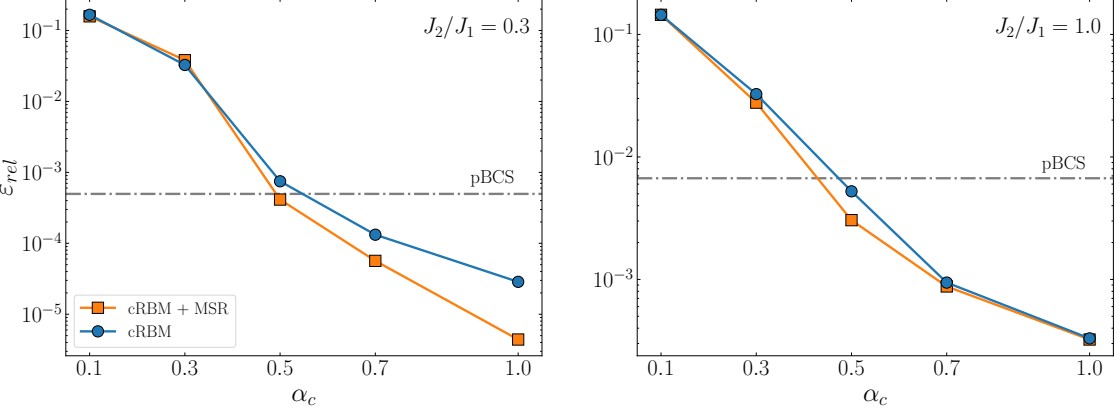

Figure 4: Accuracy of the variational energy for the $J_1 - J_2$ Heisenberg model with $N = 20$ sites, for $J_2/J_1 = 0.3$ (left panel) and $J_2/J_1 = 1$ (right panel). The results for the cRBM of Eq. (5) are reported as a function of $\alpha_c$, with and without including the Marshall signs. The accuracy of the pBCS state is also shown for comparison.

using the Stochastic Reconfiguration algorithm [46]. In all the calculations, we make use of the symmetrized RBM wave functions described above. Additionally, for ground-state calculations, we restrict our variational state to the $S_{\text{tot}}^z = \sum_R S_R^z = 0$ sector of the Hilbert space. We perform a systematic study on small clusters in which we compare the variational results achieved by RBMs with exact quantities, computed by Lanczos diagonalization. Additionally, a comparison with the variational results obtained by projected fermionic states (denoted as pBCS, see Appendix A) is reported.

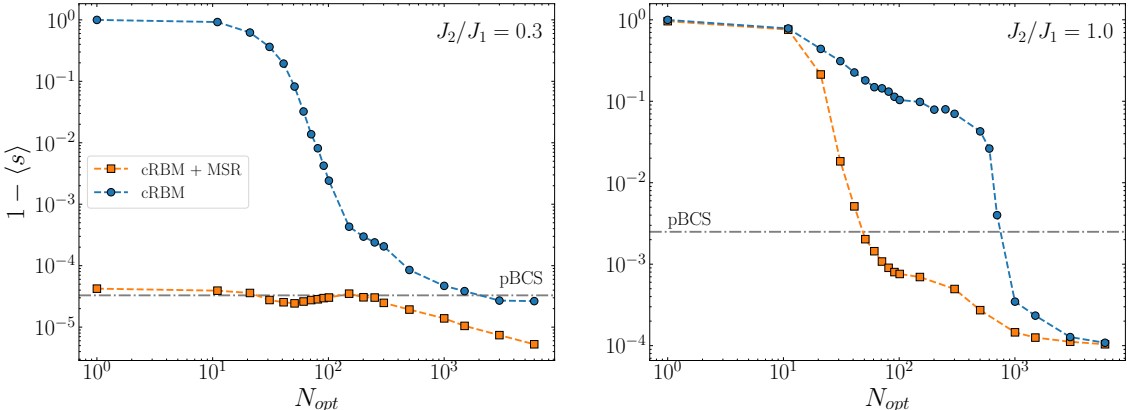

Figure 5: Evolution of the average sign defined in Eq. (9) during the optimization procedure of the cRBM state, for $J_2/J_1 = 0.3$ (left panel) and $J_2/J_1 = 1$ (right panel). Here, for each optimization step $N_{\text{opt}}$, $\langle s \rangle$ is computed (exactly) for the corresponding variational parameters.

## 3.1 Accuracy of the ground-state wave function

Let us start by comparing pmRBM and cRBM *Ansätze* on a cluster with $N = 20$ sites, for which exact results can be obtained by Lanczos diagonalization. Two values of the frustrating ratio are considered, $J_2/J_1 = 0.3$ and 1, corresponding to cases in which the Marshall-sign rule gives good and poor approximations of the exact sign structure, see Fig. 2.

In Fig. 3, we report the accuracy obtained by the pmRBM wave function for different values of $\alpha_p$, by plotting the relative error of the variational energy with respect to the exact one, namely $\varepsilon_{rel} = |(E_0 - E_{var})/E_0|$ where $E_0$ and $E_{var}$ are the exact and variational energies, respectively. We choose to consider $\alpha_m + \alpha_p = 2$, in order to fix the total number of variational parameters. The results for the cRBM state with the same number of parameters, i.e., $\alpha_c = 1$, are reported. In both cases, calculations attaching the Marshall-sign rule to the wave-function amplitudes are also considered. Without including Marshall signs, the best energy of the pm-RBM state is obtained for $\alpha_p \approx 1$, for both $J_2/J_1 = 0.3$ and 1. This means that taking the same number of variational parameters for the modulus and the phase represents the best strategy for this kind of wave function. By contrast, when including the Marshall signs, a different behavior occurs for the two values of the frustrating ratio. For $J_2/J_1 = 0.3$, where the Marshall signs represent an excellent approximation of the exact ones, the best energy of the pmRBM *Ansatz* is obtained for $\alpha_p \ll 1$; instead, for $J_2/J_1 = 1$, the optimal energy is still obtained when $\alpha_p \approx 1$. However, the lowest variational energies in Fig. 3 are those of the cRBM state. For this state, the inclusion of the Marshall-sign rule provides a substantial energy gain at $J_2/J_1 = 0.3$, while being almost ineffective for the accuracy at $J_2/J_1 = 1$. A consistent improvement with respect to pBCS wave functions [47] is achieved, even though the latter variational states require a significantly smaller number of variational parameters, e.g., up to a maximum of 6 parameters. In particular, for $J_2/J_1 = 0.3$ the energy accuracy of the cRBM is almost three orders of magnitude better than the pBCS *Ansatz*.

Having certified the better accuracy of the cRBM wave function with respect to the pmRBM state, we choose to stick to the former architecture for the remainder of the paper. In Fig. 4 we report the accuracy of the cRBM *Ansatz* when varying the network complexity $\alpha_c$. The inclusion of the Marshall-sign rule proves to be particularly effective for $J_2/J_1 = 0.3$ and $\alpha_c \gtrsim 0.5$, while being less relevant for $J_2/J_1 = 1$. Nonetheless, it is worth mentioning that the explicit inclusion of the Marshall signs always provides a computational advantage, since

it makes the optimization of the variational state easier. Indeed, let us define a measure of the difference between the phases of the cRBM wave function and the signs of the exact ground state, namely

$$\langle s \rangle = \left| \sum_{\{\sigma\}} |\Psi_0(\sigma)|^2 \text{sign}[\Psi_0(\sigma)] e^{i\Theta_{\text{cRBM}}(\sigma)} \right|, \tag{9}$$

where $\Theta_{\text{cRBM}}(\sigma) = \arg[\langle \sigma | \Psi_{\text{cRBM}} \rangle]$. As in Eq. (6), the absolute value is taken to overcome a possible global phase in the cRBM state. Then, $\langle s \rangle = 1$ whenever the phases (but not necessarily the moduli) of the cRBM state match the exact values. In Fig. 5, we track this quantity during the optimization procedure of the variational parameters, for the cases with and without the Marshall-sign rule. An evident speed-up in the convergence of the above quantity is observed when the Marshall sign structure is included, even for the case with $J_2/J_1 = 1$, for which, at the end of the simulation, no substantial energy gain is obtained by the addition of Marshall signs.

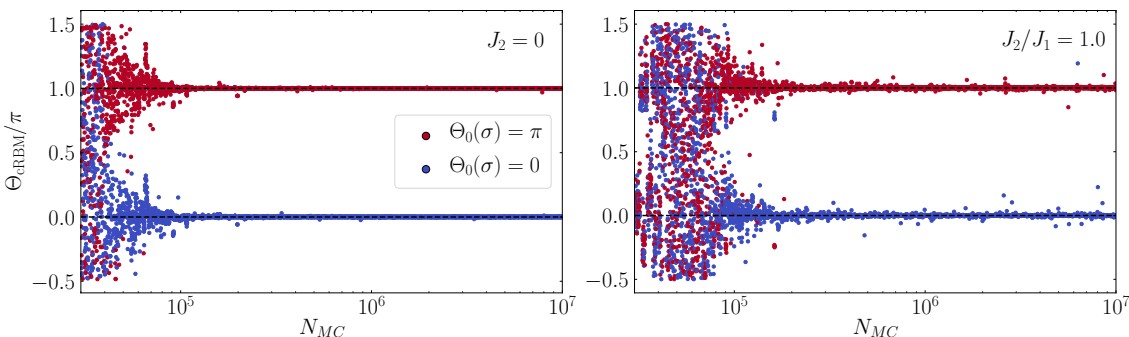

Figure 6: Evolution of the phases $\Theta_{\text{cRBM}}(\sigma)$ for the cRBM wave function with $\alpha_c = 1$ during the Monte Carlo optimization for $J_2 = 0$ (left panel) and $J_2/J_1 = 1$ (right panel). The colors of the dots denote the phase of the exact ground state wave function. The number of sites is $N = 20$. The results are plotted as a function of the number of Monte Carlo steps $N_{\text{MC}}$ and the variational parameters are updated every $10^3$ steps.

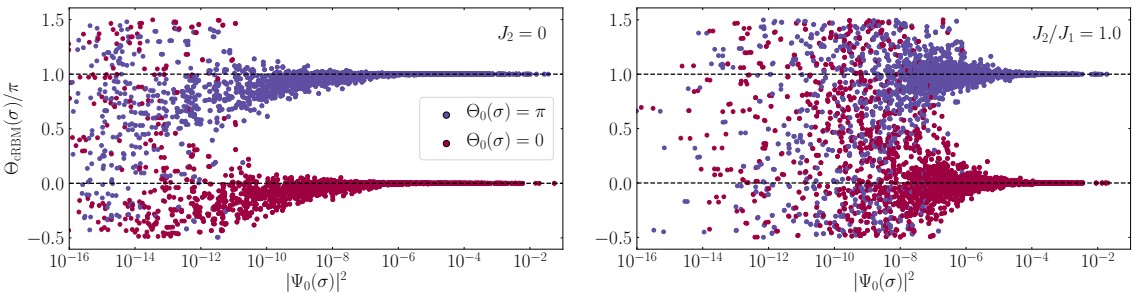

Figure 7: $\Theta_{\text{cRBM}}(\sigma)$ phases of the optimal cRBM state with $\alpha_c = 1$ for all the spin configurations with $S_{\text{tot}}^z = 0$ and momentum $k = 0$ (for $N = 20$ sites). The phases are plotted as a function of the exact weight $|\Psi_0(\sigma)|^2$ of the configurations. Results for $J_2 = 0$ (left panel) and $J_2/J_1 = 1$ (right panel) are shown. The colors of the dots denote the phase of the exact ground-state wave function.

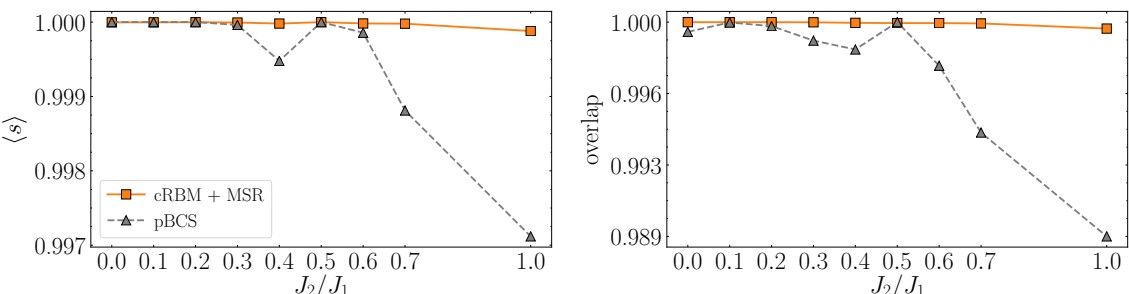

Figure 8: Average sign (9) and overlap $|\langle\Psi_0|\Psi_{\text{cRBM}}\rangle|$ for the best-energy cRBM *Ansatz* with $\alpha_c = 1$ as a function of the frustrating ratio $J_2/J_1$. The results of the pBCS state are also reported for comparison. The number of sites is $N = 20$.

Another instructive analysis of the learning process of the cRBM wave function is achieved by tracking the evolution of $\Theta_{\text{cRBM}}(\sigma)$ during the optimization procedure, computing it for the various spin configurations $|\sigma\rangle$ visited in the Monte Carlo simulation. As a benchmark, it is particularly insightful to consider the case with $J_2 = 0$, where the sign structure of the exact result is given by the Marshall-sign rule. In addition, the case with $J_2/J_1 = 1$, where the Marshall-sign rule is heavily violated, is also considered. For both cases, the values of $\Theta_0(\sigma) = \arg[\langle\sigma|\Psi_0\rangle]$ are either 0 or $\pi$, since the exact ground state is a real-valued wave function. The evolution of $\Theta_{\text{cRBM}}(\sigma)$ during optimizations is shown in Fig. 6, where blue (red) points indicate configurations for which the exact phase is $\Theta_0(\sigma) = 0$ [$\Theta_0(\sigma) = \pi$]. After an initial transient, the values of $\Theta_{\text{cRBM}}(\sigma)$ quickly converge towards the exact values. This is particularly true for $J_2 = 0$, where $\Theta_{\text{cRBM}}(\sigma)$ approaches 0 or $\pi$ with very small statistical fluctuations. A similar result is also obtained for $J_2/J_1 = 1$, even though larger fluctuations remain after convergence. It is interesting to remark that the exact signs are recovered only for the most relevant spin configurations (i.e., the ones with the largest weights), which are frequently visited in the Monte Carlo optimization, and contribute the most to the variational energy. This fact can be appreciated by looking at Fig. 7, where all the phases of the final cRBM state are shown as a function of the exact weights $|\Psi_0(\sigma)|^2$ of the corresponding spin configurations.

The results of the average sign of Eq. (9), together with the ones for the overlap between the exact ground state and the best-energy cRBM *Ansatz* $|\langle\Psi_0|\Psi_{\text{cRBM}}\rangle|$, are reported in Fig. 8 for different values of $J_2/J_1$ ($N = 20$ sites). A comparison with the results of the pBCS wave functions is also shown. We emphasize that the complex RBM always gives a better approximation of the exact ground state than the pBCS states, especially for $J_2/J_1 > 0.5$.

## 3.2 Spin-spin correlation functions

For each component $\nu = x$, $y$, and $z$ of the spin operator, we consider the expectation value of the spin-spin correlations in real space:

$$C^{\nu\nu}(r) = \frac{1}{N}\sum_R \langle\hat{S}_R^\nu\hat{S}_{R+r}^\nu\rangle\,, \tag{10}$$

and its Fourier transform in momentum space:

$$S^{\nu\nu}(k) = \sum_r e^{ikr}C^{\nu\nu}(r)\,. \tag{11}$$

Here, $\langle\cdots\rangle$ represents the expectation value over a certain quantum state. Since the RBM *Ansatz* is a function of the $z$-component of the spins only, it explicitly breaks the spin SU(2)

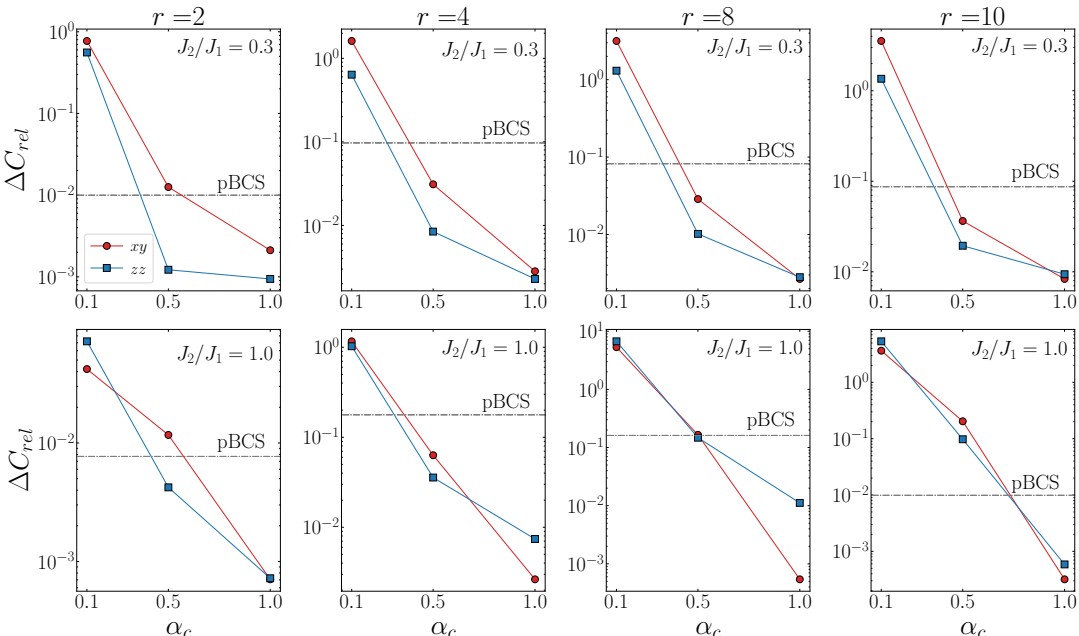

Figure 9: Relative error of the spin-spin correlation functions of the cRBM state with respect to the exact values, for $J_2/J_1 = 0.3$ (upper panels) and $J_2/J_1 = 1$ (lower panels). Results for $C^{zz}(r)$ (blue squares) and $C^{xy}(r)$ (red circles) are shown as a function of $\alpha_c$, for several distances $r$ on a $N = 20$ sites chain. The results of the (spin-isotropic) pBCS wave function are also reported for comparison.

symmetry, leading to a difference between the $z$ axis and the $x - y$ plane. However, by using a large number of variational parameters, it is possible to reduce this anisotropy and obtain almost SU(2) symmetric results. In Fig. 9, we report the relative error of the $C^{zz}(r)$ and $C^{xy}(r) = [C^{xx}(r) + C^{yy}(r)]/2$ of the cRBM state with respect to the exact spin-spin correlations, for $J_2/J_1 = 0.3$ and 1 (for $N = 20$ sites). By increasing the network complexity $\alpha_c$, the accuracy strongly improves and, consequently, also the anisotropy decreases. The pBCS wave function has SU(2) symmetry by construction and is reported for comparison. Still, its accuracy is about one order of magnitude worse than the one obtained by the best cRBM with $\alpha_c = 1$. However, it is worth remarking that the number of variational parameters is considerably different for the two classes of wave functions, with the pBCS state requiring a maximum of 6 parameters, against the 840 parameters of the cRBM *Ansatz*.

Given the tiny residual anisotropy of the cRBM *Ansatz*, we report in Fig. 10 the results for $C^{zz}(r)$ and $S^{zz}(k)$ for three representative values of the frustrating ratio, namely $J_2 = 0$ (gapless regime), $J_2/J_1 = 0.3$ (gapped regime, with commensurate spin-spin correlations), and $J_2/J_1 = 1$ (gapped regime, with incommensurate spin-spin correlations). These calculations confirm the excellent degree of approximation obtained by cRBM in all regimes. Indeed, even though the pBCS *Ansatz* also gives remarkably accurate results, the complex RBM is able to perfectly reproduce even the most challenging case with $J_2/J_1 = 1$, e.g., where the peak of $S^{zz}(k)$ is close to $k = \pi/2$.

## 3.3 Excited states

We finally report the calculations of excited states at finite momenta. Indeed, by using translational symmetry, it is possible to fix the momentum $k$ of the variational *Ansatz* in the cRBM state, see Eq. (8). In order to target the lowest-energy *triplet* excitation for each momentum,

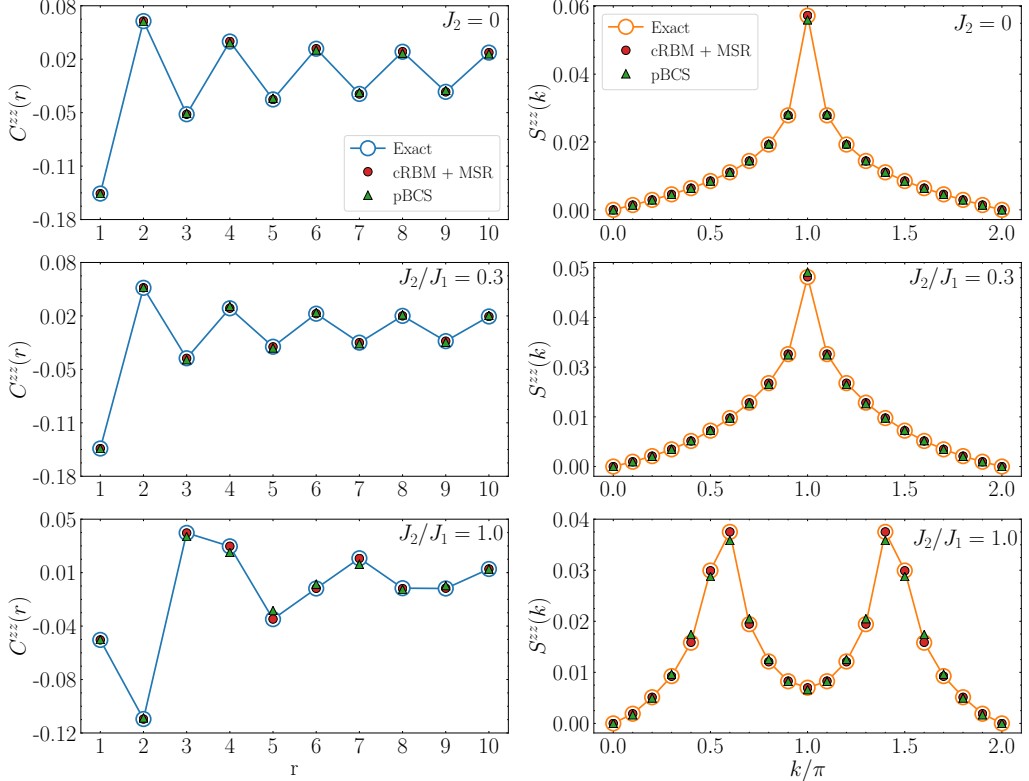

Figure 10: Spin-spin correlation function $C^{zz}(r)$ (left panels) and $S^{zz}(k)$ (right panels) for different values of $J_2/J_1$ and $N = 20$ sites. Results for the best cRBM wave function with $\alpha_c = 1$ (full circles), the pBCS state (full triangles), and the exact ground state (empty circles) are reported.

we restrict the wave function to the sector of the Hilbert space with $S^z_{\text{tot}} = 1$. The variational gaps for the lowest-lying triplets are shown in Fig. 11 for two values of the frustrating ratio in the gapped phase, $J_2/J_1 = 0.45$ and 1. The results for the gapless regime $J_2 = 0$ are perfectly compatible with the ones shown in Refs. [38, 42], and are thus not reported. The comparison of the variational energies to the exact values confirms the high accuracy of the cRBM to reproduce not only the ground-state properties, but also low-energy states.

## 4 Conclusions

In this work, we demonstrated the ability of RBM wave functions to reproduce the ground state of a frustrated spin model in one dimension, where the sign structure can be highly non-trivial (e.g., completely different from the one given by the Marshall-sign rule). The accuracy is not limited to the ground-state energy but extends to the lowest-energy triplet excitations. However, the main computational effort in achieving such accuracy is the large number of variational parameters, which grows as $O(\alpha N^2)$, where $N$ is the number of sites and $\alpha$ the complexity of the network. Hence, the optimization of the variational wave function becomes very difficult for large lattices. We emphasize the fact that, due to the fully-connected structure of the network, the transferability of the parameters when increasing the size is not possible for RBMs. By contrast, pBCS wave functions have very few variational parameters (independently of the number of spins $N$), whose optimal values rapidly converge when increasing the system

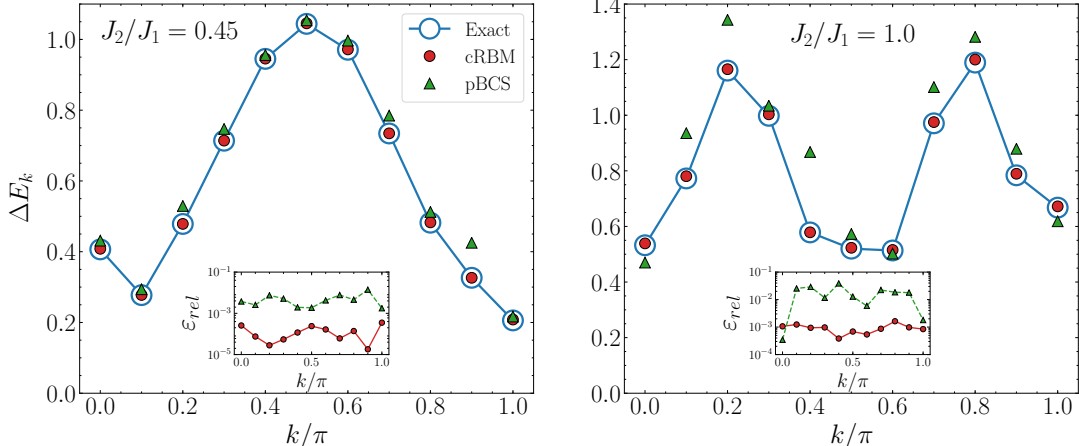

Figure 11: Lowest-energy triplet excitation of the $J_1 - J_2$ Heisenberg model for $J_2/J_1 = 0.45$ (left panel) and $J_2/J_1 = 1$ (right panel), for a chain of $N = 20$ sites. $\Delta E_k$ is the difference between the lowest triplet energy at momentum $k$ and the ground state energy. Results obtained by the best cRBM *Ansatz* with $\alpha_c = 1$ (full circles) and the pBCS state (full triangles) are shown, together with exact values (empty circles). The insets show the relative error of the variational results, i.e., $\varepsilon_{rel} = |(E_{ex,k} - E_{var,k})/E_{ex,k}|$, where $E_{ex,k}$ and $E_{var,k}$ are the exact and variational energies of the excited states, respectively.

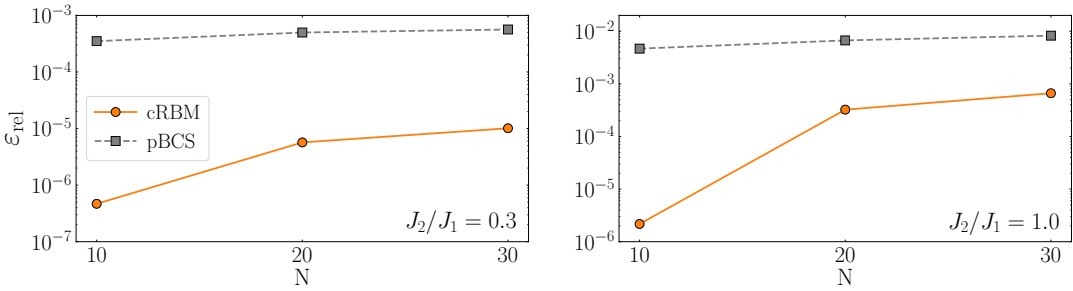

Figure 12: Size scaling of the relative error of the variational energy for the best-energy cRBM *Ansatz* with $\alpha_c = 1$. The results of the pBCS state are also reported for comparison. Calculations are done for $J_2/J_1 = 0.3$ (left panel) and $J_2/J_1 = 1$ (right panel) and $N = 10$, 20, and 30 sites.

size. Thus, the results of numerical optimizations on smaller system sizes often provide an excellent starting point for optimizations on larger lattices. Calculations with $N = 10$, 20, and 30 sites exemplify the issue of size consistency. In Fig. 12, we show the results for the relative error of the variational energy for $J_2/J_1 = 0.3$ and 1 (fixing the complexity at $\alpha_c = 1$). While the accuracy of the pBCS is lower than that of cRBMs for all sizes, pBCS states are size-consistent with very good approximation; by contrast, cRBMs with fixed complexity slightly lose accuracy when increasing the system size. As a consequence, an increase in complexity with the system size could be necessary to obtain size-consistent results. An additional remark deals with the physical interpretation of the variational states. Indeed, Gutzwiller-projected fermionic states have a transparent physical interpretation, providing a clear physical description of the phases of the system, even without computing correlation functions and observables. By contrast, RBM states lack of a physical interpretability of their variational parameters.

One possible strategy to simplify the optimization and favor a size consistent behavior could be reducing the number of parameters in the RBM state combining it with Gutzwiller-projected wave functions, e.g., using the RBM as correlator (a generalization of the standard Jastrow factor). A few works have taken this direction [27, 28], showing that with this hybrid approach it is possible to obtain very accurate results also increasing the size of the system. Other approaches focus on the generalization of the structure of the RBM network in order to improve its representational power. Some generalizations are based on the inclusion of inter-actions between hidden units of the RBM, defining the so-called *Deep* or *unrestricted Boltzmann machine*. Unfortunately, in this case tracing-out the hidden layer analytically becomes more complicated (or even impossible) [48]. Other approaches, known as *n-layer feed-forward neu-ral network* [42], rely on the inclusion of additional hidden layers to the feed-forward neural network associated to RBM (see Fig. 1). Also in this case, the possibility to have a simple analytic expression for the wave function is lost but calculations can be performed efficiently. One promising approach is to consider other classes of neural networks, such as the so-called *Convolutional Neural Networks* (CNNs) that have been shown to provide excellent results for frustrated spin systems in two dimensions [18,21–23,49]. The main advantages in using CNNs lie in sparse interactions in the network and parameters sharing [50]. Therefore, variational parameters obtained in the optimization for small lattices can be exploited as starting point for larger lattices. Taking inspiration from CNNs we can, for example, reduce the connections in the RBM, defining the so-called *Local* RBM [51]. In addition, the possibility to start the optimization on a given size with the parameters obtained on a different one highly improves the convergence; still, understanding how cutting connections influences the accuracy of the results represents an important question to investigate.

## Acknowledgments

We would like to thank Juan Carrasquilla for useful discussions.

## Appendix A   Gutzwiller-projected fermionic states

In this Appendix, we briefly describe Gutzwiller-projected fermionic wave functions, which are based upon the Abrikosov representation of spin operators [52]. Within this formalism, local spin operators are expressed in terms of fermionic operators:

$$\hat{\mathbf{S}}_R = \frac{1}{2} \sum_{\sigma,\sigma'} \hat{c}^\dagger_{R,\sigma} \tau_{\sigma,\sigma'} \hat{c}_{R,\sigma'} \,, \tag{12}$$

where $\hat{c}^\dagger_{R,\sigma}$ ($\hat{c}_{R,\sigma}$) are creation (annihilation) operators for a fermion on site $R$ and spin $\sigma$ and $\tau = (\tau_x, \tau_y, \tau_z)$ are Pauli matrices. Then, Gutzwiller-projected fermionic wave functions are constructed starting from a Bardeen-Cooper-Schrieffer (BCS) Hamiltonian:

$$\hat{\mathcal{H}}_{\text{BCS}} = \sum_{R,R',\sigma} t_{R,R'} \hat{c}^\dagger_{R,\sigma} \hat{c}_{R',\sigma} + \sum_{R,R'} \Delta_{R,R'} \left( \hat{c}^\dagger_{R,\uparrow} \hat{c}^\dagger_{R',\downarrow} + \hat{c}^\dagger_{R',\uparrow} \hat{c}^\dagger_{R,\downarrow} \right) + H.c. \,, \tag{13}$$

featuring hopping ($t_{R,R'}$) and pairing terms ($\Delta_{R,R'} = \Delta_{R',R}$). The ground-state $|\Phi_{\text{BCS}}\rangle$ of the BCS Hamiltonian is then projected into the Hilbert space of the original Heisenberg model with one electron per site (with either up or down spin):

$$|\Psi_{\text{pBCS}}\rangle = \hat{\mathcal{P}}_G |\Phi_{\text{BCS}}\rangle \,, \tag{14}$$

where $\hat{\mathcal{P}}_G = \prod_R \hat{n}_R (2 - \hat{n}_R)$ is the Gutzwiller projector defined in terms of the local electron density $\hat{n}_R = \sum_\sigma \hat{c}^\dagger_{R,\sigma} \hat{c}_{R,\sigma}$.

The parametrization of the auxiliary BCS Hamiltonian (13), i.e., the values of hopping and pairing terms, determines the properties of the variational state. Here, we use the variational *Ansätze* described in Ref. [47].

Within this framework, in addition to the pBCS *Ansatz* for the ground state (14), a variational approach to target excited states can be defined, based on Gutzwiller-projected particle-hole excitations. The procedure, outlined in Ref. [47], is employed in this work to compute the variational energies of the lowest-lying triplet excitations discussed in Section 3.3.

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
