# Peer review of "Accuracy of Restricted Boltzmann Machines for the one-dimensional $J_1-J_2$ Heisenberg model"

_SciPost Physics, doi:SciPost Phys. 12, 166 (2022)_

## Round 1 · Referee Report · Anonymous (Referee 1) · 2022-3-17

Strengths

1- Very timely 2-Well and clearly written 3- Thorough performance analysis of restricted Boltzmann machines in a well-studied example 4- Good discussion of advantages and drawbacks of RBMs and comparison to other numerical methods

Weaknesses

1-No new physics
2-No comparison with DMRG attempted here, DMRG being the state of the art method.

Report

Neural networks have emerged as a novel route to study quantum phases
of many-body systems. Here, specifically, the authors are interested in
evaluating the performance of so-called restricted Boltzmann machines (RBM) in characterizing ground-state properties of quantum magnets. Since
the sign structure of the ground state wave function of frustrated
systems hampers the applicability of QMC techniques, the authors decide
to study a one-dimensional model of frustrated magnetism, the J1-J2 chain.

The sign structure necessitates the use of complex-valued RBMs, for which
two ansaetze are compared. A fully complex network is shown to be superior.
Moreover, translational symmetry is enforced via projection operators
and the a certain sign structure can either be pre-imposed on the RBM
or not.

The authors compute variational energies, spin correlation functions
and the energies of momentum-resolved excited states. These results
are compared to those of Lanczos calculations (which can be considered
an exact benchmark) and a Gutzwiller type variational ansatz (pBCS).
The authors demonstrate good quantitative agreement of the RBM results with exact diagonalization.

They discuss the important question of scaling up the simulations to larger
systems where the considered RBM do not appear to be a promising route. The
physically motivated variational ansatz, however, can be extended to larger
systems in a straightforward manner. Alternative neural networks are identified that may be better suited for larger systems. Moreover, the variational parameters of the RBMs lack a physical interpretation. Further, the number of variational parameters for the pBCS (projected Gutzwiller) is much lower than for the RBMs.

The paper further contains very interesting results on the behavior of the
RBMs during the training phase, e.g., for the phases or the average sign.

The paper is well written and accessible, also to people who do not work
with neural networks/machine learning. It does not provide new insights into
the physics of the studied model, but the performance of the RBMs is very
carefully evaluated and discussed. It appears that RBMs are not the most
promising route for frustrated magnets. The technical details of training
the networks will presumably be of interest to machine-learning practitioners.

Overall, I conclude that this certainly very good research and publishable
science that will be of interest to its target community. The fact that RBMS are very critically evaluated is an important piece of information, follow-up work
may be triggered along the directions laid out in the conclusions sections.

Requested changes

Necessary revisionis:

1- The Gutzwiller projected wave functions are mentioned in the abstract, but neither in the abstract nor in the methods section. I may have missed it, but the acronym is never defined. The authors should a paragraph on this method in the Methods section, make sure that acronyms are properly introduced and mention the method in the introduction as well.

2- Please correct a few typos: (page 1) ".. wve functions has been defined ...", (page 14): "trasparent"

Some optional questions:

3- Could the authors make a more definite statement about the usefulness of RBMS for 2d frustrated quantum magnets?

4- The method of choice for 1D quantum magnets is still DMRG. Are there any prospects of RBMs or other neural networks becoming competitive for models of frustrated magnetism?

  • validity: top
  • significance: high
  • originality: good
  • clarity: high
  • formatting: perfect
  • grammar: good

Author:  Luciano Loris Viteritti  on 2022-04-15  [id 2388]

(in reply to Report 1 on 2022-03-17)
Category:
answer to question

We thank the referee for her/his positive report. In the following we reply to requested changes:

  1. The acronym is defined at the beginning of section 3, just before sub- section 3.1, and the definiton of Gutzwiller-projected wave functions is reported in the Appendix. The acronym is not used before its definition. The reason not to describe Gutzwiller-projected states in the body of the paper is because they are only used as a comparison for RBM states.
  2. We corrected the typos.
  3. & 4. We added a couple of sentences in the introduction to comment these points. As far as the first question is concerned, generic neural-network states can, in principle, describe quantum systems in arbitrary dimension. Howewer, from a practical point of view, higher values of the complexity α could be needed to reach the same accuracy as the one dimensional case. Concerning the second question, DMRG represents the best me- thod to solve quantum problems in one dimension, while in two or more dimensions its accuracy deteriorates. In recent years, there have been several attempts to improve both DMRG (by considering tensor-network states) and RBMs (by defining more refined architectures). Today, the latter ones have reach an accuracy that, in several cases, is competiti- ve with DMRG [see for example K. Choo, T. Neupert, G. Carleo Phys. Rev. B 100 (12), 125124 (2019)].

---

## Round 1 · Referee Report · Anonymous (Referee 2) · 2022-3-25

Report

Developing accurate variational methods is one of the central challenges in computational science and physics. Recently, Carleo and Troyer introduced variational wave functions based on artificial neural networks. Given that this research field is still in an early stage, it is an important task to perform systematic benchmark calculations to confirm the accuracy of neural network variational ansatz.

In the paper, the authors systematically investigate the accuracy of the restricted Boltzmann machine (RBM) wave function for the spin 1/2 J1-J2 Heisenberg model in one spatial dimension.

First, the authors show that complex RBM (cRBM) performs better than phase-modulus RBM (pmRBM). Then, the study focuses on the cRBM.

The authors pay attention to the sign structure of the wave function. They then show that the Marshall-sign rule helps the optimization, especially when J2 is small. The best cRBM results show better accuracy in the calculation of the ground state, both in energy and correlation functions, compared to the Gutzwiller-projected fermionic states, albeit with a much larger number of variational parameters.

By considering linear combinations of cRBMs, excited states can also be investigated (the accuracy of the ground-state is also improved). In this paper, it is shown that cRBMs give highly accurate results for excited states as well.

Finally, the authors discuss the size consistency and discuss several possibilities to improve size consistent behavior.

The paper is clearly written, and the accuracy of the RBM variational ansatz is carefully and systematically investigated. I believe that the present work is one of the important pieces of recent intensive investigations of neural-network quantum states. Thus, I recommend that this paper be accepted for publication in SciPost.

Below, I list several points (all are minor).

In Fig. 6, the definition of N_MC is not very clear. In particular, what is the difference between N_opt in Fig. 4 and N_MC in Fig. 6?

In Fig. 8, and in page 12, the overlap < Psi_0 | Psi_cRBM > should be a complex number. Do the authors show the absolute value ?

According to the inset of Fig. 11 right panel, the relative error of the pBCS at k=0 is smaller than that of the cRBM. However, looking at the right panel, the error of the pBCS appears to be considerably larger than that of the cRBM.
  • validity: -
  • significance: -
  • originality: -
  • clarity: -
  • formatting: -
  • grammar: -

Author:  Luciano Loris Viteritti  on 2022-04-15  [id 2389]

(in reply to Report 2 on 2022-03-25)

We thank the referee for her/his positive report. Here we reply to the minor points:

  1. One Monte Carlo step is obtained performing O(N) Metropolis moves (two-spin flips). With fixed variational parameters, we perform $O(10^3)$ Monte Carlo steps and then we update the variational parameters. Therefore, an optimization step corresponds to $O(N × 10^3)$ Metropolis moves. We added a sentence to clarify this point, also improving the captions of the figures.
  2. Yes, we thank the referee for pointing out this issue. We corrected it in the text.
  3. In the main panel of Fig.11, we report the variational gap, which is defined as $\Delta E_k = E_k − E_0$, where both energies are variational. In the inset, we report the accuracy of $E_k$, namely $\varepsilon_{rel}$. A lower value of $\varepsilon_{rel}$ does not necessary imply a better accuracy on the gap, because the latter depends also on the accuracy of $E_0$. At k = 0, the accuracy of the pBCS excitation is much higher than for the pBCS ground state, giving a very small $\varepsilon_{rel}$, but a relatively inaccurate gap.

---

## Round 2 · Referee Report · Anonymous (Referee 2) · 2022-4-24

Report

I have looked through the author's modifications and replies.
I recommend the paper be published in SciPost.

---

## Round 2 · Referee Report · Anonymous (Referee 1) · 2022-4-25

Strengths

see previous report

Weaknesses

see previous report

Report

The revised version is adequate for publication in SciPost Phys.

Requested changes

1- (optional) However, I still recommend that the author make an earlier mentioning of the pBCS method (i.e., in the intro) and move the contents of the Appendix before Sec. 3. Knowing what this pBCS method is essential for the reader to be able to comprehend the comparions.

---

## Round 2 · Author Response

Dear Editors,

we would like to resubmit the attached revised version of the manuscript which includes the suggestions made by the referees. We reply to the comments of the referees highlighting the changes made to the manuscript.

Sincerely yours,
Luciano Loris Viteritti, Francesco Ferrari, Federico Becca

---

## Round 2 · List of Changes

List of changes:
- we corrected the typos;
- we added a paragraph in the introduction to answer the question of referee 1 about the usefulness of RBMs for 2d frustrated systems and the comparison with DMRG approaches;
- we add a comment in the results and in the caption of Fig. 6 to clarify the difference between a N_{MC} and an N_{opt} step as required by referee 2;
- we specify that the overlap in Fig. 12 between the variational and the exact states should be understood in modulus to answer the question of referee 2.

---

## Editorial Decision

published